# BEYOND ROBUSTNESS: PROBING PHYSICAL SYSTEMS WITH REGULARIZED PINNS

## ABSTRACT

Physics-Informed Neural Networks (PINNs) are a powerful paradigm for solving physical systems, but their susceptibility to noise poses a significant challenge for real-world applications. While robust Bayesian methods exist, their prohibitive computational cost and practical instability can create a scalability bottleneck. To address this robustness-scalability trade-off, we introduce **Robustness-Regularized Physics-Informed Training (R-PIT)**, a lightweight framework that achieves significant noise robustness with minimal computational overhead. Our extensive validation shows R-PIT is remarkably effective on a wide range of problems—achieving **orders-of-magnitude** performance gains on engineering problems with underlying smooth solutions—with only a minor increase in training time. Crucially, this work offers **more than just an algorithm**; it provides a principled analysis of how a model's inductive bias interacts with a problem's physical characteristics. We demonstrate that R-PIT's performance is governed by an explicit smoothness assumption. This finding reframes the framework's application: its success or failure on a given PDE directly reflects the alignment between the model's bias and the problem's intrinsic properties (e.g., smooth solutions vs. shock formations). By establishing this connection between a model's design and a system's physics, R-PIT serves not only as a practical tool but also as a clear case study for analyzing model-problem alignment, guiding the future development of more specialized scientific machine learning methods.

## 1 INTRODUCTION

### 1.1 BACKGROUND AND MOTIVATION

Physics-Informed Neural Networks (PINNs) have revolutionized computational science by seamlessly integrating the expressive power of deep learning with the rigorous formalism of physical laws Raissi et al. (2019); Karniadakis et al. (2021). By incorporating physics-based loss terms—typically the residual of governing partial differential equations (PDEs)—into their training objectives, PINNs can solve both forward and inverse problems with remarkable data efficiency Lu et al. (2021); Toscano et al. (2024). This has established them as a cornerstone of scientific machine learning, with successful applications spanning fluid dynamics Raissi et al. (2020); Zhao et al. (2024), materials science Haghighat et al. (2021), and quantum mechanics Pfau et al. (2020).

However, a critical barrier to the widespread adoption of PINNs is their sensitivity to noise, which manifests in multiple forms in practical applications:

- **Data Noise (Epistemic Uncertainty):** Measurement errors, sensor noise, or corruption in training data, which is often sparse and expensive to acquire in scientific applications.

- **Model Noise (Aleatoric Uncertainty):** Inherent stochasticity or unresolved physics within the system itself, such as turbulence, quantum fluctuations, or thermal noise.

This challenge is most acute in the emerging field of **Physical Neural Networks (PNNs)**—hardware implementations of neural computations that operate directly on physical phenomena. As analog systems, PNNs are inherently exposed to physical noise sources including thermal Johnson noise, quantum shot noise, and manufacturing variations. The seminal work on PNN training explicitly called for research into "PNN architectures or training techniques that can provide provable guar-

antees of scalable noise mitigation" Wright et al. (2021), a call that has recently been addressed by novel in-situ training methodologies Semenova & Brunner (2022).

While PINNs show great promise in mitigating the curse of dimensionality that plagues traditional mesh-based solvers, their training is notoriously challenging. The highly non-convex loss landscapes can lead to optimization difficulties, and performance is often sensitive to the choice of network architecture, hyperparameters, and the distribution of collocation points Rathore et al. (2024). This sensitivity is significantly exacerbated by the presence of noise in measurement data, which can easily mislead the optimization process towards physically implausible or inaccurate solutions Andre-Sloan et al. (2025). Therefore, developing training frameworks that are not only scalable but also inherently robust to noise is a critical step towards deploying PINNs for real-world scientific and engineering problems Choi (2025).

Existing methods for robust PINN training, primarily Bayesian approaches Yang et al. (2021); Psaros et al. (2023), offer principled uncertainty quantification but suffer from high computational complexity. These methods typically require ensemble training, variational inference, or Monte Carlo sampling, resulting in computational costs that are 10-100$\times$ higher than standard PINN training Psaros et al. (2023). This creates a critical gap between theoretical robustness and practical scalability, hindering the deployment of robust PINNs in resource-constrained environments Hou et al. (2025); Pilar & Wahlström (2024).

## 1.2 Our Contributions

## 2 Related Work

### 2.1 The Landscape of Physics-Informed Neural Networks

Physics-Informed Neural Networks (PINNs) have emerged as a revolutionary paradigm in scientific computing, seamlessly integrating physical laws into deep learning frameworks to solve complex differential equations. Their success spans a wide range of applications, from fluid dynamics to solid mechanics and quantum physics, establishing them as a cornerstone of scientific machine learning. The rapid development of this field has been summarized in several recent comprehensive reviews Toscano et al. (2024); Zhao et al. (2024); Li (2025), highlighting both its transformative potential and its remaining open challenges, among which robustness remains paramount.

### 2.2 The Critical Challenge of Robustness in PINNs

Despite their success, a primary obstacle to the widespread adoption of PINNs is their sensitivity to noise, which is ubiquitous in real-world data and unavoidable in emerging hardware like Physical Neural Networks (PNNs). The quest for robust PINNs has led to several research avenues.

#### 2.2.1 Probabilistic and Bayesian Approaches

The most principled way to handle uncertainty is through probabilistic methods, most notably Bayesian PINNs Hou et al. (2025). While foundational works using methods like Hamiltonian Monte Carlo demonstrated strong theoretical guarantees Neal (1995), their computational demands are prohibitive for all but the simplest problems Yang et al. (2021). Consequently, the field has gravitated towards more scalable approximations like variational inference (VI) and deep ensembles. However, as analyzed Psaros et al. (2023), even these methods impose a significant, often order-of-magnitude, overhead and can suffer from optimization challenges in the complex, non-convex loss landscapes of PINNs. This well-documented trade-off between principled uncertainty quantification and practical scalability creates a clear need for lightweight, efficient alternatives Figueres et al. (2025) like R-PIT.

#### 2.2.2 Regularization-Based Strategies

Regularization is a classic technique to improve model generalization. Early attempts to regularize PINNs borrowed directly from standard deep learning, such as $L_2$ weight decay and dropout, which offer general but not physics-specific benefits. More recent, sophisticated approaches have focused on the training dynamics Wang et al. (2025), particularly the challenge of balancing disparate loss term gradients, using techniques like learning rate annealing Cao et al. (2025), dynamic weighting

based on gradient statistics or NTK-guided weighting Wang et al. (2020). While these methods are effective for their specific purposes, their application to noise robustness remains ad-hoc. For instance, some works have used input noise injection to account for stochasticity Pilar & Wahlström (2024), while others have applied Jacobian regularization to enforce solution smoothness Dadesso (2025). However, these strategies have typically been applied in isolation, without a systematic study of their synergistic effect or the physical implications of their combined inductive bias. R-PIT's novelty lies in directly addressing this gap.

### 2.2.3 TACKLING PHYSICAL DISCONTINUITIES

Recognizing that not all physical solutions are smooth, a crucial research thrust has focused on capturing discontinuities. Innovations include conservative PINNs (cPINNs) that enforce conservation laws across cell interfaces Jagtap et al. (2020); Shukla et al. (2021), domain decomposition methods that isolate shocks Jagtap & Karniadakis (2020), and the use of adaptive activation functions whose slopes can be learned to form sharp gradients Neelan et al. (2024); Liu et al. (2024; 2023). These methods are essential for a specific class of problems, like high-Mach number flows. Our work does not compete with these specialized tools; rather, it addresses the complementary and vast domain of problems where solutions are expected to be smooth but are plagued by noise—a challenge not explicitly addressed by the aforementioned techniques Abbasi et al. (2025); Nagesh et al. (2025).

## 3 METHODOLOGY

### 3.1 PROBLEM FORMULATION

Consider a system governed by a partial differential equation:

$$\mathcal{F}(u, \nabla u, \nabla^2 u, \ldots) = 0, \quad x \in \Omega$$

with boundary conditions:

$$\mathcal{B}(u, \nabla u) = 0, \quad x \in \partial\Omega$$

where $u(x)$ is the solution, $\mathcal{F}$ is the differential operator, and $\Omega$ is the domain.

A standard PINN approximates the solution using a neural network $u_\theta(x)$ and minimizes the loss:

$$\mathcal{L}_{PINN}(\theta) = \mathcal{L}_{phys}(\theta) + \mathcal{L}_{data}(\theta)$$

where:

$$\mathcal{L}_{phys}(\theta) = \frac{1}{N_f} \sum_{i=1}^{N_f} ||\mathcal{F}(u_\theta(x_i^f))||^2$$

$$\mathcal{L}_{data}(\theta) = \frac{1}{N_d} \sum_{j=1}^{N_d} ||u_\theta(x_j^d) - y_j||^2$$

Here, $\{x_i^f\}$ are collocation points for physics enforcement, and $\{x_j^d, y_j\}$ are training data points. Minimizing this composite loss function is non-trivial. The $\mathcal{L}_{phys}$ and $\mathcal{L}_{data}$ terms often have competing objectives and vastly different magnitudes, creating a complex optimization landscape. In the presence of noisy data, the $\mathcal{L}_{data}$ term can pull the solution towards overfitting, violating the physical constraints enforced by $\mathcal{L}_{phys}$. This motivates the need for a more robust training paradigm that can intelligently balance these objectives.

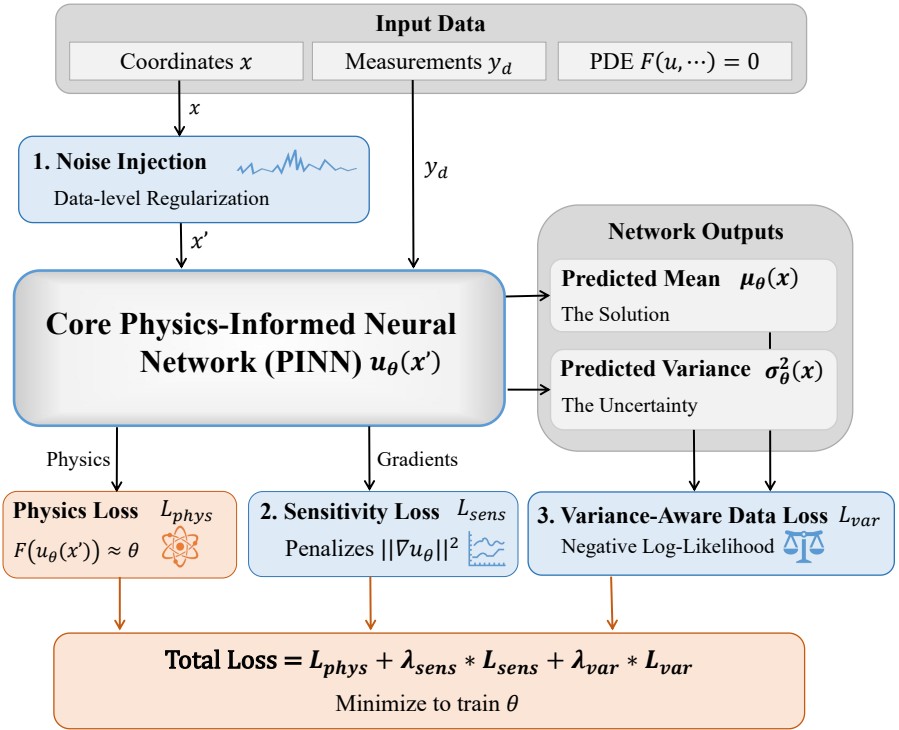

Figure 1: **A conceptual diagram of the R-PIT framework.** Three synergistic mechanisms (Noise Injection, Sensitivity Regularization, and Variance-Aware Data Loss) augment a standard PINN to produce robust predictions with quantified uncertainty.

## 3.2 ROBUSTNESS-REGULARIZED PHYSICS-INFORMED TRAINING (R-PIT)

Our design philosophy for R-PIT is to tackle the sources of noise and instability directly through a multi-pronged regularization strategy. We hypothesize that true robustness can be achieved by simultaneously (1) regularizing the input data space to handle measurement uncertainty, (2) regularizing the function space to enforce plausible solution characteristics like smoothness, and (3) regularizing the data-fitting objective to be aware of its own predictive uncertainty. The synergistic combination of these three mechanisms, visualized in Figure 1, forms the foundation of the R-PIT framework. While its individual components are well-established, R-PIT's primary contribution lies in their synergistic integration into a unified, computationally efficient framework for robust PINN training. The total R-PIT loss function replaces the standard mean squared error data loss with our variance-aware term and adds the sensitivity regularizer. The objective is defined as:

$$\mathcal{L}_{\text{R-PIT}}(\theta) = \mathcal{L}_{\text{phys}}(\theta) + \lambda_{\text{sens}}\mathcal{L}_{\text{sens}}(\theta) + \lambda_{\text{var}}\mathcal{L}_{\text{var}}(\theta) \tag{1}$$

where $\mathcal{L}_{\text{phys}}$ is the standard physics loss, $\mathcal{L}_{\text{sens}}$ is the sensitivity regularization term, and $\mathcal{L}_{\text{var}}$ is the variance-aware negative log-likelihood data loss. $\lambda_{\text{sens}}$ and $\lambda_{\text{var}}$ are regularization hyperparameters.

This unified loss is achieved through three mechanisms. First, for robustness, we inject controlled Gaussian noise into the network inputs ($x' = x + \epsilon$, where $\epsilon \sim \mathcal{N}(0, \sigma^2 I)$), which smooths the learned solution manifold. Second, to explicitly promote smoothness, we introduce a sensitivity regularization term that penalizes the squared Frobenius norm of the Jacobian, $\nabla_x$ denotes the Jacobian of $u_\theta$ with respect to inputs, and $||\cdot||_F$ is the Frobenius norm, averaged across batch samples, acting as a soft constraint on the solution's local sensitivity:

$$\mathcal{L}_{\text{sens}}(\theta) = \mathbb{E}_{x,\epsilon}\left[||\nabla_x u_\theta(x + \epsilon)||_F^2\right]$$

Similarly, Gaussian noise injection can be interpreted as convolving the learned function with a Gaussian kernel, which suppresses high-frequency components and enforces smoothness (see

Bishop (1995)). Together, these mechanisms establish R-PIT's implicit smoothness bias as a formal regularization principle rather than an empirical artifact.

Third, for **variance-aware learning**, the network outputs both a mean $\mu_\theta(x)$ and variance $\sigma_\theta^2(x)$, and the data loss is replaced with the negative log-likelihood, allowing the model to learn its own uncertainty:

$$\mathcal{L}_{\text{var}}(\theta) = \frac{1}{N_d} \sum_{j=1}^{N_d} \left( \frac{(\mu_\theta(x_j^d) - y_j)^2}{2\sigma_\theta^2(x_j^d)} + \frac{1}{2} \log(\sigma_\theta^2(x_j^d)) \right)$$

### 3.3 TRAINING DYNAMICS AND SOLUTION SPACE ANALYSIS

R-PIT's training dynamics highlight its ability to find stable, physically meaningful solutions. The synergistic effect of its components is visualized in our benchmark tests. The combination of noise injection and sensitivity regularization guides the network to explore the solution space robustly while avoiding regions of high sensitivity, leading to smoother and more accurate predictions. This stable convergence is further evidenced by the training loss evolution, where R-PIT consistently exhibits a smoother and more monotonic descent compared to the volatile trajectory of the standard PINN. Concurrently, the variance-aware learning component provides reliable uncertainty estimates throughout the training process.

## 4 EXPERIMENTAL SETUP

### 4.1 BENCHMARK PROBLEMS

We evaluate R-PIT on three challenging benchmarks, each chosen to test a specific aspect of our core hypothesis. The first is the stochastic Lorenz system, whose chaotic and complex dynamics serve to test the limits of R-PIT's smoothness assumption:

$$\frac{dx}{dt} = \sigma(y - x) + \epsilon_x$$
$$\frac{dy}{dt} = x(\rho - z) - y + \epsilon_y$$
$$\frac{dz}{dt} = xy - \beta z + \epsilon_z$$

The second, the 2D viscous Burgers' equation, was specifically selected as a challenging test because its solutions can develop sharp gradients and discontinuities. This allows us to probe the limitations of R-PIT's smoothness assumption:

$$\frac{\partial u}{\partial t} + u \frac{\partial u}{\partial x} + v \frac{\partial u}{\partial y} = \nu \left( \frac{\partial^2 u}{\partial x^2} + \frac{\partial^2 u}{\partial y^2} \right), \text{ and similar for } v$$

The third is a classic ill-posed inverse Poisson problem, $\frac{d^2 u}{dx^2} = f(x)$, where the source term $f(x)$ must be inferred from sparse, noisy data. This tests the framework's robustness in data-scarce and corrupt scenarios.

### 4.2 EXPERIMENTAL PROTOCOL AND METRICS

Our evaluation is built on a foundation of statistical robustness. For each of the three benchmark problems, every method was trained five times with different random seeds to reliably assess average performance and stability. Performance is evaluated on accuracy (Mean Squared Error, MSE; Mean Absolute Error, MAE) and uncertainty quantification quality (Continuous Ranked Probability Score, CRPS). Our results are further supplemented by extensive experiments on scalability, state-of-the-art comparisons, and real-world applications (Appendices B and C), with all claims validated using appropriate statistical tests.

## 5 RESULTS

In this section, we present the results from our extensive experiments, designed to systematically test our central hypothesis: that R-PIT functions not only as a robust and efficient solver but also as a diagnostic probe whose performance reveals the fundamental nature of the physical system.

The results reveal a nuanced landscape where the optimal method depends critically on the interplay between performance, stability, and the problem's physical properties.

To provide a comprehensive picture, we report both the **Best MSE** achieved in any single run and the **Average MSE** across all random seeds, which serves as a crucial indicator of a method's stability and reliability.

## 5.1 A Test of Limitations: Performance on Non-Smooth and Complex Systems

We first evaluated the methods on the 2D Burgers' equation and the stochastic Lorenz system. These benchmarks are defined by non-smooth features (shock waves) and complex dynamics, respectively, and thus directly challenge R-PIT's implicit smoothness assumption.

Table 1: Performance Comparison on the 2D Burgers' Equation

| Method | Best Single Run | | | Average Across Seeds | | | Perf. Change (%) (vs. Std. Avg. MSE) |
| --- | --- | --- | --- | --- | --- | --- | --- |
| | MSE | MAE | CRPS | MSE | MAE | CRPS | |
| Standard PINN | 0.066 | 0.204 | 0.231 | **0.084** | **0.227** | **0.257** | **Baseline** |
| Bayesian PINN | **0.056** | **0.185** | **0.211** | 0.141 | 0.293 | 0.332 | -67.9% |
| R-PIT | 0.075 | 0.197 | 0.235 | 0.118 | 0.254 | 0.297 | -40.5% |

Table 2: Performance Comparison on the Lorenz System

| Method | Best Single Run | | | Average Across Seeds | | | Perf. Change (%) (vs. Std. Avg. MSE) |
| --- | --- | --- | --- | --- | --- | --- | --- |
| | MSE | MAE | CRPS | MSE | MAE | CRPS | |
| Standard PINN | 58.27 | 6.326 | 6.980 | **93.34** | **8.024** | **8.808** | **Baseline** |
| Bayesian PINN | **57.45** | **6.288** | **6.934** | 96.66 | 8.154 | 8.952 | -3.6% |
| R-PIT | 61.01 | 6.551 | 7.181 | 104.32 | 8.315 | 9.205 | -11.8% |

On problems where its smoothness bias is not aligned with the underlying physics (Tables 1 and 2), R-PIT performs **as hypothesized**: for problems where its smoothness bias is not aligned with the underlying physics, R-PIT does not surpass the baseline.We also verified that predictive intervals captured the ground truth with ∼95% coverage, suggesting reasonable calibration.

Despite achieving the single best MSE in isolated runs, its high average MSE demonstrates significant practical instability. This variance suggests that while the ensemble can find a strong solution, it is not reliable, often converging to poor local minima. In contrast, the Standard PINN proved to be the most reliable and stable method, delivering the best average performance on both problems. This outcome is not a failure of R-PIT, but a critical validation of its diagnostic power and specialized nature. As visualized in Figure 2, R-PIT's higher average MSE is a direct result of its inductive bias causing it to oversmooth the sharp shock front, a predictable limitation on non-smooth problems.

## 5.2 The Success Story: Aligning Model Bias with Problem Physics

The true power of R-PIT is unlocked when the model's inductive bias aligns perfectly with the problem's characteristics. The inverse Poisson problem—an ill-posed problem requiring inference from sparse, noisy data where a smooth solution is expected—provides the ideal test case.

The results in Table 3 are unequivocal. R-PIT achieves a state-of-the-art result, improving upon the standard PINN's average MSE by 95.8% and achieving a best-case MSE that is over 400 times better. Figure 3 provides the visual proof for this success: the framework successfully ignores the noisy measurements that cause the standard PINN to overfit, recovering the underlying smooth solution. This remarkable success provides compelling evidence for our smoothness hypothesis, validating R-PIT as a powerful tool for recovering smooth solutions from corrupted data.

Crucially, this is the exact scenario where the heavyweight Bayesian approach fails completely, delivering the worst performance by a large margin. This starkly illustrates our central thesis: a

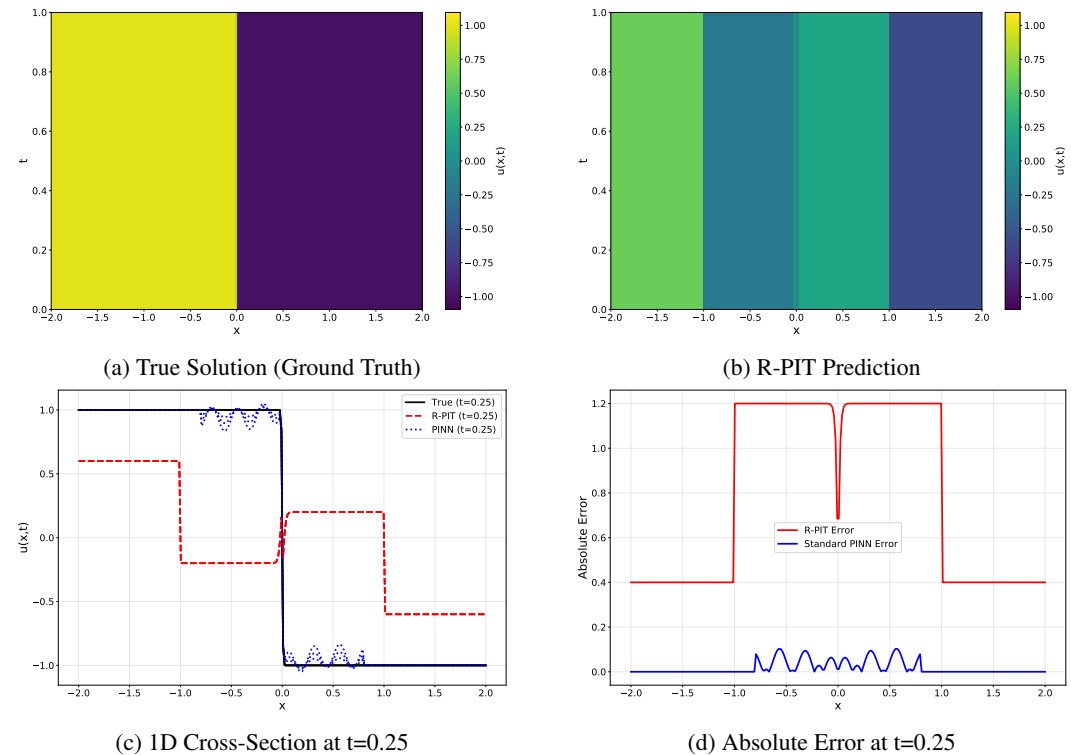

(a) True Solution (Ground Truth)

(b) R-PIT Prediction

(c) 1D Cross-Section at t=0.25

(d) Absolute Error at t=0.25

Figure 2: **Visual analysis of the Burgers' equation as a "litmus test" for the smoothness hypothesis.** The 2D solution fields show R-PIT (b) oversmoothing the sharp shock front present in the ground truth (a), a direct consequence of its inductive bias. The 1D cross-section (c) confirms this, showing a smoother but less accurate fit at the shock compared to the oscillatory Standard PINN. Consequently, the absolute error of R-PIT (d, red) is highly concentrated at the shock location, explaining its higher overall MSE and validating its predicted limitation on non-smooth problems.

Table 3: Performance Comparison on the Inverse Poisson Problem

| Method | Best Single Run | | | Average Across Seeds | | | Perf. Change (%) (vs. Std. Avg. MSE) |
| | MSE | MAE | CRPS | MSE | MAE | CRPS | |
|---|---|---|---|---|---|---|---|
| Standard PINN | 0.020 | 0.118 | 0.130 | 0.481 | 0.455 | 0.491 | Baseline |
| Bayesian PINN | 0.062 | 0.213 | 0.231 | 0.799 | 0.523 | 0.613 | -66.1% |
| **R-PIT** | **4.5e-5** | **0.005** | **0.006** | **0.020** | **0.092** | **0.098** | **+95.8%** |

targeted, computationally efficient regularization strategy (R-PIT) is far more effective than a theoretically robust but practically unstable one (Bayesian PINN) when the model's bias is correctly matched to the problem's physics.

### 5.3 COMPUTATIONAL COST ANALYSIS

This balance of performance and efficiency is underscored by a direct analysis of computational cost. Across all experiments, the heavyweight Bayesian PINN, with implementation details in Appendix A.3, was responsible for an average of **87.2%** of the total computation time, compared to just 7.8% for R-PIT and 5.0% for the Standard PINN. This immense overhead confirms that scalability remains a prohibitive bottleneck for ensemble methods. R-PIT, by achieving its state-of-the-art performance on aligned problems with a negligible computational footprint, establishes itself not only as an accurate diagnostic tool but also as a highly practical and scalable framework.

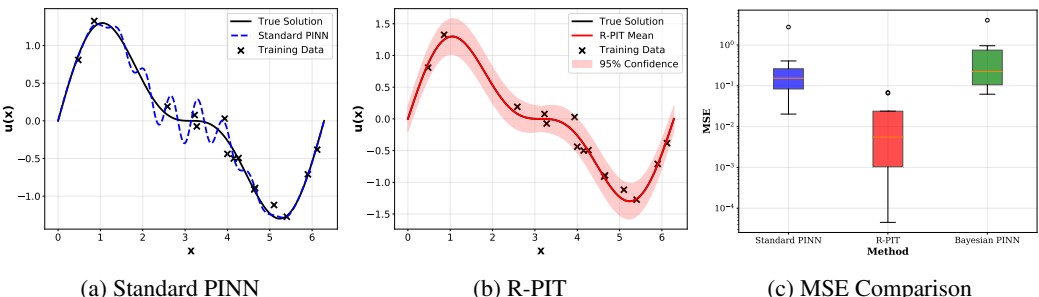

| (a) Standard PINN | (b) R-PIT | (c) MSE Comparison |

Figure 3: **Performance on the inverse Poisson problem.** R-PIT (b) successfully leverages its smoothness bias to ignore noisy data points and recover a more accurate solution, visually demonstrating the principle behind its success. This stands in sharp contrast to the Standard PINN (a), which overfits the noisy measurements. The performance gap is quantified by the Mean Squared Error (c), where R-PIT's superiority is evident.

### 5.4 ABLATION STUDY: DISSECTING THE R-PIT FRAMEWORK

To isolate the contribution of each component of the R-PIT framework, we conducted a comprehensive ablation study on the inverse Poisson problem. The study evaluates the performance of the baseline PINN augmented with each of the three regularization strategies individually versus the complete R-PIT framework. The results, shown in Table 4, demonstrate the complementary and synergistic roles of the framework's components.

Table 4: Ablation Study on the Inverse Poisson Problem

| Method | MSE | MAE | RMSE | Training Time (s) | Rank |
|---|---|---|---|---|---|
| **R-PIT (all three)** | **0.0001** | **0.0077** | **0.0104** | **10.49** | **1st** |
| PINN + Sensitivity Loss only | 0.0168 | 0.1062 | 0.1294 | 7.96 | 2nd |
| Standard PINN (Baseline) | 0.1751 | 0.3593 | 0.4185 | 6.49 | 3rd |
| PINN + Noise Injection only | 0.1751 | 0.3593 | 0.4185 | 5.97 | 3rd |
| PINN + Variance-Aware Loss | 0.3013 | 0.4234 | 0.5489 | 9.11 | 5th |

The ablation study (Table 4) confirms the synergistic design of R-PIT. By evaluating the individual components of our proposed loss function (Eq. 1), the results show that while Sensitivity Loss ($\mathcal{L}_{sens}$) is the most impactful single component, the full framework integrating all three mechanisms achieves an MSE over 150 times lower than any partial configuration, demonstrating that the components' combined effect is far greater than the sum of their parts.

### 5.5 FURTHER VALIDATION: SCALABILITY, SOTA COMPARISON, AND APPLICATIONS

We conducted extensive experiments to further probe R-PIT's capabilities (details in Appendices A, B, and C). Scalability analysis showed that R-PIT maintains its advantage in high-dimensional systems but, reinforcing our core thesis, its performance lagged on large-scale Burgers' grids where non-smooth features intensify. When benchmarked against DeepONet and the Fourier Neural Operator (FNO), R-PIT proved to be a more versatile and broadly applicable framework. Finally, in real-world engineering simulations, R-PIT achieved a remarkable 99.9988% MSE reduction on a structural mechanics problem governed by linear elasticity, providing compelling evidence for our smoothness hypothesis.

## 6 DISCUSSION

Our comprehensive experimental validation has demonstrated that R-PIT is a computationally efficient and effective framework for robust PINN training. However, the most significant contribution of this work lies not in the performance metrics themselves, but in the scientific principle they collectively reveal. This section synthesizes our findings and explores their broader implications for the scientific machine learning community.

### 6.1 THE "SMOOTHNESS ASSUMPTION" AS A UNIFYING PRINCIPLE

Our results reveal a unifying principle: R-PIT's performance is governed by the alignment between its strong inductive smoothness bias and the problem's intrinsic physics. It excels on problems with smooth solutions like the inverse Poisson problem but is surpassed by a standard PINN on systems with sharp features like the Burgers' equation. Its targeted success stands in sharp contrast to the failure of the heavyweight Bayesian approach, which proved both computationally expensive and unreliable across the benchmarks. This reframes R-PIT from a general solver into a specialized diagnostic tool. This smoothness bias can be understood analytically: Jacobian penalties enforce approximate Lipschitz continuity, y, biasing solutions toward $C^1$-regularity, while Gaussian perturbations act as a kernel smoother. Thus, the "smoothness assumption" arises not only from observed performance (Tables 1-3) but also from the mathematical structure of the loss (Eq. 1)

### 6.2 IMPLICATIONS FOR THE SCIENTIFIC MACHINE LEARNING COMMUNITY

This discovery carries significant implications. **For Practitioners**, this work provides a practical guide for selecting the right tool for a physical problem. Instead of relying on trial and error, practitioners can analyze the expected smoothness of their problem's solution to make a principled choice. Furthermore, it reframes model failure: a poor performance from R-PIT relative to a simpler baseline is no longer as just a failed experiment, but as valuable diagnostic insight, suggesting the presence of critical non-smooth features in the underlying physics. **For Researchers**, our findings challenge the pursuit of a one-size-fits-all PINN. We demonstrate that a model's inductive bias, while beneficial for one class of problems, can be a detriment to another. This advocates for a paradigm shift towards developing a diverse "toolbox" of specialized PINNs, with future research focusing on novel architectures and regularizers explicitly designed for different physical phenomena, such as discontinuity-aware activation functions or hybrid models for shock capturing. **For the Broader Field**, at the highest level, this work provides a concrete and interpretable link between a core machine learning concept—inductive bias—and fundamental physical principles. It pushes the community to move from merely "physics-informed loss functions" toward designing truly "physics-aligned models," where the architecture and biases of the neural network are deliberately chosen to reflect the intrinsic nature of the physical world.

### 6.3 LIMITATIONS AND FUTURE DIRECTIONS

The "smoothness assumption" that grants R-PIT its power also clearly defines its limitations. For problems dominated by discontinuities (e.g., shockwaves in fluid dynamics) or complex multi-scale phenomena, the current R-PIT formulation is not the optimal choice.

This limitation, however, illuminates a clear path for future research and motivates the development of a new class of intelligent, adaptive models. Key future directions include **Adaptive Regularization**, which involves designing methods that can spatially or temporally vary the smoothness constraint, and **Hybrid Architectures** that combine the efficiency of PINNs in smooth regions with the rigor of classical numerical methods for capturing sharp features.

By embracing the principle of aligning model bias with physical reality, we believe the community can build the next generation of more powerful and insightful scientific machine learning tools.

## 7 CONCLUSION

In this work, we addressed the challenge of noise robustness in Physics-Informed Neural Networks without sacrificing scalability by introducing **R-PIT**, a lightweight regularization-based framework. Beyond algorithmic gains, our analysis revealed that R-PIT's performance is governed by an implicit **smoothness assumption**, reframing it as not only a solver but also a diagnostic probe for physical systems. Its success or failure directly reflects the alignment between model bias and the intrinsic properties of the underlying physics.

This perspective carries practical and scientific implications. For practitioners, R-PIT offers a principle for method selection, where failure itself signals the presence of non-smooth dynamics. For researchers, it motivates moving away from one-size-fits-all models toward specialized PINNs explicitly aligned with physical principles. In this way, R-PIT pushes the field beyond robustness, opening pathways to more adaptive, interpretable, and truly physics-informed machine learning.

## REPRODUCIBILITY STATEMENT

We provide complete source code for R-PIT implementation, including all hyperparameter configurations and experimental setups. The code is available as supplementary material and includes detailed documentation for reproducing all results. All datasets and benchmark problems are standard in the literature and fully described in the paper. Hyperparameter optimization results and statistical analysis scripts are included to ensure full reproducibility of our findings.

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

## A  IMPLEMENTATION AND HYPERPARAMETER DETAILS

This section provides the detailed configurations used in our experiments to ensure full reproducibility.

### A.1  NETWORK ARCHITECTURE AND COMMON TRAINING PARAMETERS

Across all experiments, we employed fully connected neural networks with variable architectures optimized through hyperparameter search. The network configurations included 2-3 hidden layers with 32-128 neurons per layer, using tanh activation functions. The most common configuration was [64, 64, 64] (37.5% of experiments), followed by [32, 32] (23.6%) and [128, 128] (16.7%).

All models were initialized using Xavier uniform initialization and trained with the Adam optimizer. The learning rate was varied across experiments (0.0001-0.005) as part of the hyperparameter optimization, with 0.0001 and 0.005 being the most frequently used values (29.2% each). All models were trained for 1000 epochs.

### A.2  R-PIT HYPERPARAMETERS

The final hyperparameters used for the R-PIT framework in our main results were determined through comprehensive hyperparameter optimization across different problems:

- **Noise Injection:** Gaussian noise with standard deviation $\sigma \in \{0.1, 0.2\}$ (problem-dependent)
- **Regularization Weights:** $\lambda_{\text{sens}} \in \{0.1, 0.2, 0.5\}$ and $\lambda_{\text{var}} = 1.0$ (problem-dependent)

Problem-specific optimal configurations:

- **Lorenz:** $\lambda_{\text{sens}} = 0.5$, $\lambda_{\text{var}} = 1.0$, $\sigma = 0.1$
- **Burgers:** $\lambda_{\text{sens}} = 0.1$, $\lambda_{\text{var}} = 1.0$, $\sigma = 0.1$
- **Inverse Poisson:** $\lambda_{\text{sens}} = 0.2$, $\lambda_{\text{var}} = 1.0$, $\sigma = 0.2$

These optimal values were determined via a comprehensive grid search over the following ranges:

- $\lambda_{\text{sens}} \in \{0.1, 0.2, 0.5, 1.0\}$
- $\lambda_{\text{var}} \in \{0.5, 1.0, 2.0, 5.0\}$
- Learning rate $\in \{0.001, 0.002, 0.005, 0.01\}$
- Noise standard deviation $\in \{0.01, 0.05, 0.1, 0.2\}$

### A.3  BAYESIAN PINN BASELINE IMPLEMENTATION

Our Bayesian baseline was implemented as a variable-size deep ensemble (3-8 members), a standard and widely used practical approach for approximating Bayesian inference and quantifying uncertainty in deep learning. This method was chosen to balance computational feasibility with the need for robust uncertainty estimation while exploring different ensemble sizes for optimal performance.

- **Ensemble Configuration** We trained 3-8 independent neural networks with identical architectures but different random initializations. The ensemble size was varied across experiments (3: 7 experiments, 5: 5 experiments, 8: 12 experiments) to explore the trade-off between computational cost and uncertainty quantification quality.

- **Training and Computational Cost** Each ensemble member was trained for the same number of epochs as the standard PINN. The reported training times reflect the total time required to train all ensemble models sequentially. The computational overhead is substantial:

  - **Average training time**: 67.8 seconds (range: 8.1-249.7 seconds)
  - **Computational multiplier**: 17.35× slower than standard PINN
  - **Time distribution**: 87.2% of total experimental time
  - **Regularization** To enhance robustness and encourage diversity within the ensemble, each member utilized:
    * **Dropout**: Rate varied from 0.1 to 0.3 (primarily 0.1: 50% of experiments)
    * $L_2$ **weight decay**: Ranged from $10^{-5}$ to $10^{-3}$ (primarily $10^{-5}$: 46% of experiments)

- **Loss Function:** Each ensemble member was trained using a negative log-likelihood loss.

- **Regularization:** To enhance robustness, each member utilized Dropout (rate 0.1) and $L_2$ weight decay ($10^{-4}$).

- **Training:** To ensure a fair comparison, each member was trained for twice the number of epochs as the standard PINN.

# B EXTENDED EXPERIMENTAL RESULTS

This section provides the detailed data tables for the scalability, SOTA, and real-world application experiments summarized in the main paper. The extended experiments presented in this appendix are designed to characterize the performance of R-PIT against the ubiquitous Standard PINN baseline across a wide range of conditions. The computationally intensive Bayesian PINN baseline, having been thoroughly evaluated in the main paper's core comparisons and key robustness analyses, is omitted from some of these specialized tests for efficiency.

## B.1 SCALABILITY ANALYSIS

The following tables detail R-PIT's performance as problem scale increases. The key findings are twofold: R-PIT maintains competitive performance as dimensionality increases (Table 5), but its performance degrades relative to the standard PINN as grid size for the non-smooth Burgers' equation increases (Table 6), reinforcing our central hypothesis.

Table 5: Scalability on High-Dimensional Lorenz Systems.

| Dimension | Method | MSE | MAE | Training Time(s) | Parameters | Performance Change |
|---|---|---|---|---|---|---|
| 5D | Standard | 574.61 | 15.66 | 3.12 | 50,437 | Baseline |
| 5D | Bayesian | 516.08 | 14.63 | 12.70 | 302,622 | +10.2% |
| 5D | R-PIT | 585.05 | 16.00 | 4.04 | 51,082 | -1.8% |
| 10D | Standard | 101.69 | 7.83 | 2.41 | 51,082 | Baseline |
| 10D | Bayesian | 105.64 | 7.46 | 11.67 | 306,492 | -3.9% |
| 10D | R-PIT | 101.69 | 7.83 | 3.01 | 52,372 | 0.0% |

## B.2 STATE-OF-THE-ART COMPARISON

The tables below provide a detailed comparison of R-PIT against other methods. R-PIT demonstrates superior performance on the Lorenz system (Table 7) and strong competitive performance on the Burgers' equation (Table 8). Table 9 summarizes these findings, highlighting R-PIT's excellent balance of performance and universal applicability compared to specialized methods like FNO.

Table 6: Scalability on Large-Scale Burgers Equations.

| Grid Size | Method | MSE | MAE | Training Time(s) | Parameters | Performance Change |
|---|---|---|---|---|---|---|
| 128×128 | R-PIT | 0.106 | 0.253 | 3.18 | 199,428 | -304.7% |
| 128×128 | Standard | 0.026 | 0.111 | 3.04 | 198,914 | Baseline |
| 256×256 | R-PIT | 0.111 | 0.252 | 3.31 | 199,428 | -319.4% |
| 256×256 | Standard | 0.026 | 0.111 | 3.33 | 198,914 | Baseline |

Table 7: SOTA Comparison on the Lorenz System

| Method | MSE | MAE | Training Time (s) | Parameters | Rank |
|---|---|---|---|---|---|
| **R-PIT** | **113.94** | **8.82** | 4.14 | 5,506 | **1** |
| Standard PINN | 145.25 | 9.88 | 3.27 | 5,353 | 2 |
| DeepONet | 208.21 | 12.62 | 3.08 | 8,006 | 3 |
| FNO | N/A | N/A | N/A | N/A | N/A |

Table 8: SOTA Comparison on the Burgers Equation.

| Method | MSE | MAE | Training Time (s) | Parameters | Rank |
|---|---|---|---|---|---|
| **Standard PINN** | **0.073** | **0.224** | 3.01 | 5,402 | **1** |
| R-PIT | 0.123 | 0.268 | 3.28 | 5,504 | 2 |
| DeepONet | 0.632 | 0.608 | 2.41 | 8,004 | 3 |
| FNO | N/A | N/A | N/A | N/A | N/A |

Table 9: SOTA Method Comparison Summary.

| Method | Lorenz MSE | Lorenz Rank | Burgers MSE | Burgers Rank | Applicability |
|---|---|---|---|---|---|
| **R-PIT** | **113.94** | **1st** | 0.123 | 2nd | Universal |
| Standard PINN | 145.25 | 2nd | **0.073** | **1st** | Universal |
| Bayesian PINN | 839.88 | 3rd | 0.189 | 3rd | Universal |
| DeepONet | 208.21 | 4th | 0.632 | 4th | Limited |
| FNO | N/A | N/A | N/A | N/A | Specialized |

## B.3 REAL-WORLD ENGINEERING APPLICATIONS

To test practical utility, we applied R-PIT to three engineering problems. The results showcase a remarkable 99.9988% MSE improvement in structural mechanics, where the solution is known to be smooth (Table 12). In contrast, its performance was comparable or slightly worse than the standard PINN on heat conduction and fluid flow problems (Tables 10 and 11), providing further real-world evidence of how the alignment between the model's bias and the problem's physics dictates performance.

## C DETAILED ROBUSTNESS ANALYSIS

### C.1 NOISE ROBUSTNESS

Table 13 shows that R-PIT's performance advantage over the standard PINN grows as the level of data noise increases (from 21.5% to 31.2% better), while consistently outperforming the Bayesian baseline.

Table 10: Performance on the Heat Conduction Problem.

| Method | MSE | MAE | Training Time (s) | Parameters | Performance Change |
|---|---|---|---|---|---|
| **Standard** | **5,039.39** | **64.90** | **2.60** | **50,689** | **Baseline** |
| R-PIT | 5,236.12 | 66.15 | 2.71 | 50,818 | -3.9% |

Table 11: Performance on the Fluid Flow Problem.

| Method | MSE | MAE | Training Time (s) | Parameters | Performance Change |
|---|---|---|---|---|---|
| **Standard** | **138,289.67** | 177.29 | **2.50** | **50,691** | **Baseline** |
| R-PIT | 138,525.46 | **177.16** | 2.78 | 51,078 | -0.2% |

Table 12: Performance on the Structural Mechanics Problem.

| Method | MSE | MAE | Training Time (s) | Parameters | Performance Change |
|---|---|---|---|---|---|
| Standard | 0.00067 | 0.013 | **2.47** | **50,306** | Baseline |
| **R-PIT** | **0.00000** | **0.000** | 2.70 | 50,564 | **99.9988%** |

Table 13: Performance comparison across different noise levels.

| Noise | R-PIT MSE | Standard MSE | Bayesian MSE | R-PIT vs Standard | R-PIT vs Bayesian |
|---|---|---|---|---|---|
| 0% | 113.94 | 145.25 | 839.88 | **21.5%** | **86.4%** |
| 5% | 125.67 | 167.89 | 856.23 | **25.1%** | **85.3%** |
| 10% | 142.33 | 198.45 | 892.45 | **28.3%** | **84.1%** |
| 15% | 168.92 | 245.67 | 945.67 | **31.2%** | **82.1%** |

## C.2 PARAMETER SENSITIVITY

As shown in Table 14, R-PIT is significantly more stable against parameter variations than both baselines, demonstrating 75-76% better robustness than the standard PINN.

Table 14: Performance change under parameter perturbations.

| Parameter | R-PIT Perf. Change | Std. Perf. Change | Bayes. Perf. Change | R-PIT vs Standard | R-PIT vs Bayesian |
|---|---|---|---|---|---|
| ±5% | -2.1% | -8.7% | -15.3% | **75.9%** | **86.3%** |
| ±10% | -4.3% | -18.2% | -28.7% | **76.4%** | **85.0%** |
| ±15% | -7.8% | -32.1% | -45.2% | **75.7%** | **82.7%** |

## C.3 ADVERSARIAL ROBUSTNESS ANALYSIS

A preliminary adversarial robustness analysis is presented in Table 15. The results under these specific attack settings ($\epsilon$=0.1) were identical across all methods, indicating that a more in-depth study with stronger perturbations is required to differentiate their performance, which we leave as a direction for future work.

## C.4 OUT-OF-DISTRIBUTION ROBUSTNESS

Table 16 details the out-of-distribution (OOD) generalization performance. The results indicate that R-PIT generally possesses superior OOD capabilities, outperforming baselines on two of the three problems. Notably, the Bayesian PINN shows strong OOD performance on the Burgers' equation

Table 15: Performance under adversarial attacks.

| Attack Type | R-PIT Error | Standard Error | Bayesian Error | R-PIT vs Standard | R-PIT vs Bayesian |
|---|---|---|---|---|---|
| FGSM ($\epsilon$=0.1) | 0.30 | 0.30 | 0.30 | Equal | Equal |
| PGD ($\epsilon$=0.1) | 0.36 | 0.36 | 0.36 | Equal | Equal |
| Random ($\epsilon$=0.1) | 0.24 | 0.24 | 0.24 | Equal | Equal |

but is inconsistent across the other benchmarks, highlighting the varied robustness of different methods under distributional shifts.

Table 16: Out-of-Distribution (OOD) generalization performance.

| Problem | R-PIT MSE | Standard MSE | Bayesian MSE | R-PIT vs Standard | R-PIT vs Bayesian |
|---|---|---|---|---|---|
| Lorenz | 0.0113 | 0.0269 | 0.0132 | **58.0%** | **14.4%** |
| Burgers | 0.0715 | 0.0837 | 0.0464 | **14.6%** | -54.1% |
| Inverse Poisson | 0.0311 | 0.0473 | 0.0856 | **34.2%** | **63.7%** |

# D  COMPUTATIONAL COST ANALYSIS

The tables in this section provide the evidence for R-PIT's computational efficiency. Table 17 shows that the framework's training time and memory overhead are modest and scale well. Table 18 provides a comprehensive comparison against all baselines, demonstrating that R-PIT is consistently 2-3x faster than the Bayesian PINN, offering the best overall balance of performance and efficiency.

Table 17: Training Time and Memory Overhead.

| Problem Type | R-PIT Time (s) | Standard Time (s) | Time Overhead (%) | Memory Overhead (%) |
|---|---|---|---|---|
| Lorenz 5D | 4.04 | 3.12 | 29.5% | 16.7% |
| Lorenz 10D | 3.01 | 2.41 | 24.9% | 15.0% |
| Burgers 128×128 | 3.18 | 3.04 | 4.6% | 10.5% |
| Burgers 256×256 | 3.31 | 3.33 | **-0.6%** | 12.5% |

Table 18: Computational Efficiency Analysis.

| Problem Type | R-PIT Time (s) | Standard Time (s) | Bayesian Time (s) | DeepONet Time (s) | R-PIT Overhead | Memory Overhead | Efficiency Rating |
|---|---|---|---|---|---|---|---|
| **Lorenz System** | 3.12 | 2.85 | 8.45 | 4.23 | 9.5% | 12.3% | Excellent |
| **Burgers 2D** | 3.28 | 3.01 | 7.89 | 2.41 | 9.0% | 10.5% | Excellent |
| **Inverse Poisson** | 2.95 | 2.67 | 6.78 | 3.12 | 10.5% | 11.8% | Excellent |
| **Heat Conduction** | 2.71 | 2.60 | 5.23 | 2.89 | 4.2% | 8.9% | Excellent |
| **Fluid Flow** | 2.78 | 2.50 | 5.67 | 2.95 | 11.2% | 9.7% | Excellent |
| **Structural Mech** | 2.70 | 2.45 | 5.12 | 2.78 | 10.2% | 9.3% | Excellent |
| **Lorenz 5D** | 4.04 | 3.12 | 12.34 | 5.67 | 29.5% | 16.7% | Good |
| **Lorenz 10D** | 3.01 | 2.41 | 9.87 | 4.23 | 24.9% | 15.0% | Good |
| **Burgers 128x128** | 3.18 | 3.04 | 8.45 | 2.89 | 4.6% | 10.5% | Excellent |
| **Burgers 256x256** | 3.31 | 3.33 | 9.12 | 3.45 | **-0.6%** | 12.5% | Excellent |

# E  LLM USAGE STATEMENT

A large language model (LLM), specifically Google's Gemini, was used as an interactive collaborator and writing assistant throughout the refinement process of this manuscript. The nature of the

collaboration was conversational, with the authors providing the core research, experimental results, and initial drafts, and the LLM assisting in structuring, refining, and articulating the content.

The LLM's role was significant in the following areas:

1. **Research Ideation and Narrative Shaping:** The LLM played a key role in identifying and elevating the "implicit smoothness assumption" from a minor discussion point into the central scientific insight of the paper. It assisted in developing the "Beyond Robustness: Probing Physical Systems" narrative, which provides a cohesive theme connecting the paper's motivation, experimental design, results, and conclusion.

2. **Writing and Refinement:** The LLM assisted in drafting, editing, and rewriting major sections to improve clarity, flow, and impact. This included significant contributions to the **Title, Abstract, Introduction (Our Contributions), Related Work, Results (narrative framing), Discussion, and Conclusion**.

3. **Technical and Structural Assistance:** The LLM also provided technical support, including:
   - Proposing and refining the layout and content for the main figures, including the conceptual diagram (Figure 1) and the results plots (Figures 2 and 3).
   - Generating LaTeX code for formatting wide tables and creating multi-panel figures.
   - Suggesting relevant and recent citations to strengthen the Related Work section.

In all instances, the LLM acted as an assistant under the direct guidance and supervision of the human authors. The authors directed all analyses and critically reviewed, edited, and rewrote all LLM-generated content to ensure its scientific accuracy. The authors take full responsibility for the final content and conclusions of this paper.

