# OpenReview forum: "Beyond Robustness: Probing Physical Systems with Regularized PINNs"
_ICLR.cc/2026/Conference — ICLR 2026 Conference Withdrawn Submission_

### Official Review · Reviewer_xdPT · 2025-10-20

**Soundness:** 2
**Presentation:** 2
**Contribution:** 1
**Rating:** 2
**Confidence:** 4

**Summary:**

The manuscript proposes a novel training scheme for PINNs, which combines noise injection for inputs, a smoothness constraint, and a variance-aware data loss. It is shown that the trained PINN performs superior on the Poisson equation.

**Strengths:**

Improving PINN training is an important research direction, with many papers developing targeted solutions for concrete applications. This paper, which aims for a generic approach based on a smoothness assumption, is a welcome addition to this field of literature. The paper further shows several examples where this approach is promising.

**Weaknesses:**

The paper currently has a few weaknesses which prevent me from recommending acceptance. A major concern is that the paper seems to be written largely by AI, which the authors acknowledge, but which leads to overly redundant/enthusiastic statements that just don't read well in a scientific paper. Also, there are several inconsistencies (e.g., the central hypothesis at the beginning of Sec. 5 was not stated before). This being a problem the scientific community has to face in general, I will now move to the technical concerns:
- Several modeling aspects are not clear. For example, how is the variance trained for the variance-aware data loss? Is there any physics-information used, or is this quantity trained just from data?
- Several claims are not supported by evidence: For example, in Sec. 3.3 it is stated that the variance-aware learning component provides reliable uncertainty estimates, but this is not shown in the main part of the paper, neither are there references to the appendix. The CRPS is not used in the main paper. How are the bad outcomes in Table 1 and 2 "a critical validation of [R-PIT's] diagnostic power"? The failure on the Burgers' equation is not due to shocks, because R-PIT is certainly capable of modeling them -- they are just placed at the wrong positions ($x$ -1 and 1, respectively). Is there another explanation? Also, in reference to Sec. 6.2, how can we be sure that R-PIT performs well on all smooth problems, if only a very small set of experiments was made?
- Experimental details are missing: What are the parameters of $\epsilon$ in the Lorenz system, what is $f(x)$ for the Poisson problem? How is the training data generated for the Poisson problem?
- The experimental evidence is weak, since the R-PIT seems to be optimized over hyperparameters while the competing methods were not optimized in that regard. Further, why are two failure modes shown, when the appendix contains several other successful examples? Why should we expect poor performance on the stochastic Lorenz system -- is this system non-smooth? If it is smooth, then should we not expect good performance via the inductive bias of R-PIT? Further, can we not achieve robustness against noise in the Poisson problem simply by downscaling the data term in the PINN? Since R-PIT has two loss weight terms, the loss weights of the standard PINN terms should be tuned as well.
- What does adversarial robustness actually mean in the context of PINNs?

Minor issues:
- incorrect usage of citet vs. citep.
- epistemic and aleatoric uncertainty appear to be confused
- the mentioning of PNNs seems off-topic in the intro
- Section 1.2 is missing

**Questions:**

My comment on the paper's weaknesses contains several questions, I would appreciate if the authors could address them. For example:
- How is the variance trained for the variance-aware data loss? Is there any physics-information used, or is this quantity trained just from data?
- How are the bad outcomes in Table 1 and 2 "a critical validation of [R-PIT's] diagnostic power"?
- The failure on the Burgers' equation is not due to shocks, because R-PIT is certainly capable of modeling them -- they are just placed at the wrong positions ($x$ -1 and 1, respectively). Is there another explanation?
- In reference to Sec. 6.2, how can we be sure that R-PIT performs well on all smooth problems, if only a very small set of experiments was made?
- What are the parameters of $\epsilon$ in the Lorenz system, what is $f(x)$ for the Poisson problem?
- How is the training data generated for the Poisson problem?
- Why are two failure modes shown, when the appendix contains several other successful examples?
- Why should we expect poor performance on the stochastic Lorenz system -- is this system non-smooth? If it is smooth, then should we not expect good performance via the inductive bias of R-PIT?
- Can we not achieve robustness against noise in the Poisson problem simply by downscaling the data term in the PINN? Since R-PIT has two loss weight terms, the loss weights of the standard PINN terms should be tuned as well.
- What does adversarial robustness actually mean in the context of PINNs?

**Details Of Ethics Concerns:**

Maybe the overly obvious usage of LLMs could be seen problematic. The authors acknowledge that in Appendix E, but I think that such heavy use could qualify as "unprofessional" or bordering "research integrity".

---

### Official Review · Reviewer_HdCo · 2025-10-30

**Soundness:** 2
**Presentation:** 2
**Contribution:** 2
**Rating:** 4
**Confidence:** 5

**Summary:**

This paper proposes Robustness-Regularized Physics-Informed Training (R-PIT), as a framework that improves the robustness and stability of Physics-Informed Neural Networks (PINNs) to noise through three mechanisms, 1. input noise injection, 2. sensitivity regularization and 3. variance aware data loss for UQ.
The authors argue that these mechanisms collectively impose an implicit smoothness bias, aligning model behavior with smooth physical systems. Experiments on Lorenz, Burgers, and inverse Poisson problems support their claim that R-PIT excels when the physical system’s solution is smooth and underperforms when discontinuities dominate. The paper reframes robustness as a manifestation of model–problem alignment, rather than as an isolated optimization property.
The authors, however, use inflated claims that are not adequately supported, theoretically or experimentally.  Standard benchmark experiments are conducted and this remains one of my problems with PINNs developments as using them against realworld problems often does not deliver results/advantages as expected/advertised.  This paper attempts to handle some of the known issues but only sparingly demonstrate on realworld problems (structural mechanics).  Lorenz (chaotic), Burgers (shock-prone), and inverse Poisson (smooth) are canonical PDE testbeds, not real-world industrial systems.  the three “engineering” problems — heat conduction, fluid flow, and structural mechanics mentioned are not based on field data, results are on numerically simulated toy systems derived from standard PDEs with known analytic or finite-element solutions.
In terms of generalizations and extensions my assessment is the following:
1.Domain diversity is moderate
2.Noise realism is low
3.Scalability evidence is good (Tables 17-18)
4.Empirical generalization is poor
5.Robustness to domain shift is weak


The use and narrative contribution of LLMs in writing and ideation are disclosed but might raise minor concerns over original authorship or insight, though the core scientific contributions are clearly attributed to the authors

**Strengths:**

R-PIT is a  lightweight framework integrating three regularization mechanisms: noise injection, sensitivity regularization, and variance-aware data loss.
R-PIT framing moves beyond traditional robustness to introduce the idea of physics-aligned inductive bias.  While these 3 components are individually new, their synergistic integration and interpretation through an “implicit smoothness bias” provides an original contribution. The idea of treating the model’s inductive bias as a diagnostic probe for physical systems reframes PINNs not merely as solvers but as model–problem alignment tools.

Writing style and clarity: The manuscript is clear and well-organized, with schematics and side-by-side comparisons that effectively illustrate the argument. The writing style however is sometimes overly rhetorical, with marketing-like phrasing (“the true power of R-PIT is unlocked,” “remarkable success,” etc.

Relevance: The study addresses a central and unresolved challenge in physics-informed machine learning: balancing robustness to noise with computational scalability. R-PIT is lightweight and practical the proposed regularization adds minimal computational overhead (≈8%) per results presented.

Accuracy Checks: The methodology and reported results are internally consistent. The mathematical formulation of R-PIT and its loss decomposition (Eq. 1) aligns with established principles of noise-based regularization (Bishop, 1995).
Results include empirical studies over multiple PDEs, ablations, scalability, robustness, and real-world benchmarks. The smoothness assumption is clearly analyzed both empirically and theoretically.

Inclusion of code reproducibility, full hyperparameter details and and LLM authorship statement improve transparency and reproducibility

**Weaknesses:**

While the paper is generally well written, there are several areas of concern that must be addressed, these concerns include the loose use of terminology and overreach in claims adding to confusion.  For example,
The paper uses both PINN and PNN, primarily in Section 1.1. The rest of the paper focuses almost entirely on PINNs, and R-PIT is explicitly a training framework for PINNs. PNNs are mentioned only in the introduction to motivate hardware noise robustness as a use-case where such methods might ultimately matter.  By invoking PNNs in the motivation, the authors implicitly broaden their claim to hardware robustness, but no hardware-level or analog experiments are presented. Thus, the inclusion feels rhetorical rather than scientific.
Epistemic uncertainty is mentioned but the definition is inaccurate, sensor noise is used in this category, but Sensor or measurement noise is not epistemic, it is typically considered aleatoric (irreducible)
Overstated quantitative claims: Phrases such as “orders-of-magnitude improvement” are not consistent across benchmarks.
The authors use Inflated writing tone, e.g., “true power of R-PIT” and Orders of magnitude (not quantified) detract from scientific and neutral writing.

Other issues are:
There is very limited theoretical formalism. The smoothness principle is qualitatively reasoned but lacks formal convergence proofs or bias–variance trade-off derivation.
The discussion on probabilistic baseline uses ensemble Bayesian PINNs only and omits comparison to modern probabilistic or Laplace PINNs.
while implied in claims, there is no adaptive or discontinuity-aware variant tested, though suggested in future work.
while Statistical robustness is addressed via multi-seed averaging and CRPS metrics the paper would benefit from uncertainty bounds on performance metrics and visual comparison across multiple noise levels, not just selected instances
The claim that R-PIT acts as a “diagnostic tool” is conceptually supported but could be empirically strengthened by showing how its failure correlates with a known smoothness transition in PDEs.

**Questions:**

Recommendations:
Model Noise (Aleatoric Uncertainty) -- I would not call this model noise
PNN vs PINN
The claim that R-PIT acts as a “diagnostic tool” is conceptually supported but could be empirically strengthened by showing how its failure correlates with a known smoothness transition in PDEs.
while Statistical robustness is addressed via multi-seed averaging and CRPS metrics the paper would benefit from uncertainty bounds on performance metrics and visual comparison across multiple noise levels, not just selected instances
The claim that R-PIT acts as a “diagnostic tool” is conceptually supported but could be empirically strengthened by showing how its failure correlates with a known smoothness transition in PDEs.

Questions:
Claim: R-PIT “probes” a system’s physics through its inductive bias. How would one formally quantify or measure alignment between model bias and the PDE’s intrinsic smoothness?
Can this concept of alignment be generalized beyond smoothness—e.g., to conservation or symmetry properties?
Have you looked into a theoretical justification to explain why R-PIT’s smoothness bias improves robustness but fails near discontinuities? Could R-PIT be modified to adaptively relax smoothness (e.g., spatially weighted Jacobian penalties) near shocks?
The paper appears to invert standard definitions of epistemic and aleatoric uncertainty. How do you reconcile your terminology with Bayesian UQ conventions (e.g., Kendall & Gal, 2017)?

Typos etc.
line 53:  The seminal work on PNN training explicitly ... cite the work closer to point of relevance.
use neutral tones in writing

---

### Official Review · Reviewer_dc9R · 2025-11-01

**Soundness:** 3
**Presentation:** 1
**Contribution:** 2
**Rating:** 2
**Confidence:** 3

**Summary:**

The paper proposes and demonstrates a training recipe called Robustness Regularized Physics-Informed Training (R-PIT for PINNs that adds 3 things:
- a little noise on the inputs,
- a Jacobian penalty on how sensitive the network is to the inputs ,
- a data loss that learns per-point variance (so noisy data is down-weighted).
It empirically works well when the true solution is smooth (e.g., inverse Poisson with noisy, sparse data), but it tends to over-smooth and do worse on problems with shocks or sharp features like Burgers, Lorenz.

I do think the paper is not ready for acceptance in the current stage, in terms of its magnitude of contribution, clarity and completness, correctness of writing.

**Strengths:**

- The method is simple with clear, simple objective that’s easy to reproduce and tune just two weights and a noise level.
- The method achieves strong results where smoothness holds and large gains on inverse Poisson with noisy sparse data.
- The author are honest to the adv/disadv or their method and explicitly state the smoothness assumption and treats failure as a diagnostic signal for non-smooth physics.

**Weaknesses:**

- Writing quality and technical clarity are major issues. The paper is difficult to follow and appears to be missing content. For instance, the “Contributions” section has only a header with no details, and the definitions of aleatoric vs. epistemic uncertainty are reversed and never used afterward, which is confusing. This undermines the trustfulness on this paper. Aleatoric should be inherent/random noise in the data or process (e.g., measurement noise, sensor noise) and epistemic uncertainty should be from lack of knowledge/model misspecification and can in principle be reduced with more data and better modeling.
- The proposed method hass limited breadth of superiority, it clear stands out mainly on inverse Poisson while perform worse at Burgers/Lorenz than standard PINN.
- The contribution is incremental and integrative as each ingredient (noise, Jacobian control, heteroscedastic NLL) is established in previous work and the contribution focuses on packaging and framing rather than a new core mechanism.
- The fundamental to the method’s bias has risk of over-smoothing on discontinuities is and no adaptive mechanism is provided.

**Questions:**

- Can you give simple rules of thumb for when the smoothness bias improves generalization?

- Can this method be adaptive? Could the noise level or sensitivity weight vary across space/time to avoid washing out shocks?

- Uncertainty quality: How well-calibrated is the learned variance across tasks (any calibration curves)?

- Hyperparameter sensitivity: How fragile are results to the choice of noise level and penalty weights?

---

### Official Review · Reviewer_tHJy · 2025-11-03

**Soundness:** 2
**Presentation:** 3
**Contribution:** 2
**Rating:** 2
**Confidence:** 5

**Summary:**

The paper proposes a version of physics informed neural network that is robust to noises in data. The key idea is twofold: (1) add a sensitivity regularizer, namely, penalizing the gradient norm of the PINN with noise injected, and (2) let the PINN outputs a Gaussian distribution, including both mean and predictive variance.

**Strengths:**

1. Motivation is reasonable and clear
2. The method is reasonable

**Weaknesses:**

1. The novelty is very weak. The idea of predicting both mean and variance --- to form a Gaussian distribution, and then fit these Gaussian distribution to the data likelihood is a standard approach for training probabilistic neural network. For example [1] and [2]. The idea of penalizing the loss with gradient norm (sensitivity) is not new as well. It is known in the community that such regularization can help with robustness and generalization, see [3] and [4]. The paper is simply a straightforward combination of the two known methods in the training of PINNs.

2. The additional cost over training an Bayesian PINN is improperly exaggerated. If you use SGLD, SGHMC,  variance inference with stochastic training and SWA-Gaussian (SWAG), the cost is comparable to optimizing the loss with SGD for point estimation, because they all rely on stochastic gradient. If you use deep ensemble, you need to train the PINN in parallel multiple times. If you use Laplace method, you just need one more step for block Hessian computation. If you use Monte Carlo dropout, there is no difference in the cost with SGD. All these approaches do not lead to prohibitive,  orders-of-magnitude cost increase, yet are working at least in certainty scenarios. There is not an accurate and comprehensive discussion about the cost of different Bayesian training method for neural networks.

3. The comparison is very limited. It only compares one version of Bayesian PINN --- but there can be many other choices --- and ablation versions of the proposed method. There is no comparison in running time, which is claimed to be a major advantage of the proposed method. It is a bit disappointed that R-PIT does not exhibit advantage over standard PINN and the selected Bayesian PINN in Table 1 and Table 2. It only shows improvement in Table 3. The empirical results therefore are not strong enough to demonstrate the effectiveness of the proposed method.




[1] Lakshminarayanan B, Pritzel A, Blundell C. Simple and scalable predictive uncertainty estimation using deep ensembles[J]. Advances in neural information processing systems, 2017, 30.
[2] Nix D A, Weigend A S. Estimating the mean and variance of the target probability distribution[C]//Proceedings of 1994 ieee international conference on neural networks (ICNN'94). IEEE, 1994, 1: 55-60.
[3] Nagarajan V, Kolter J Z. Gradient descent GAN optimization is locally stable[J]. Advances in neural information processing systems, 2017, 30.
[4] Zhao Y, Zhang H, Hu X. Penalizing gradient norm for efficiently improving generalization in deep learning[C]//International conference on machine learning. PMLR, 2022: 26982-26992.

**Questions:**

see above

---

### Note · Authors · 2025-11-12

**Comment:**

After careful consideration of the reviewers’ thoughtful and detailed feedback, I have decided to withdraw this submission to make substantial improvements to the work. I sincerely thank all reviewers for their valuable comments and suggestions, which will guide a more rigorous theoretical and empirical revision of the paper for future submission.

**Withdrawal Confirmation:**

I have read and agree with the venue's withdrawal policy on behalf of myself and my co-authors.